# Cross-Disciplinary Genomics Approaches to Studying Emerging Fungal Infections

**DOI:** 10.3390/life10120315

**Published:** 2020-11-28

**Authors:** Pria N. Ghosh, Lola M. Brookes, Hannah M. Edwards, Matthew C. Fisher, Phillip Jervis, Dana Kappel, Thomas R. Sewell, Jennifer M.G. Shelton, Emily Skelly, Johanna L. Rhodes

**Affiliations:** 1Department of Infectious Disease Epidemiology, MRC Centre for Global Infectious Disease Analysis, St Mary’s Campus, Imperial College London, London W2 1PG, UK; Lola.Brookes@ioz.ac.uk (L.M.B.); hannah.edwards11@imperial.ac.uk (H.M.E.); matthew.fisher@imperial.ac.uk (M.C.F.); uccapaj@ucl.ac.uk (P.J.); dana.kappel19@imperial.ac.uk (D.K.); t.sewell@imperial.ac.uk (T.R.S.); j.shelton@imperial.ac.uk (J.M.G.S.); e.skelly@imperial.ac.uk (E.S.); johanna.rhodes@imperial.ac.uk (J.L.R.); 2Unit for Environmental Sciences and Management, North-West University, Potchefstroom 2520, South Africa; 3Institute of Zoology, Zoological Society of London, London NW1 4RY, UK; 4Royal Veterinary College, Hawkshead Lane, North Mymms, Herts AL9 7TA, UK; 5Department of Chemistry, University College London, London WC1H 0AJ, UK; 6UK Centre for Ecology & Hydrology, Wallingford OX10 8BB, UK

**Keywords:** cross-disciplinary, fungal-omics, emerging fungal pathogen

## Abstract

Emerging fungal pathogens pose a serious, global and growing threat to food supply systems, wild ecosystems, and human health. However, historic chronic underinvestment in their research has resulted in a limited understanding of their epidemiology relative to bacterial and viral pathogens. Therefore, the untargeted nature of genomics and, more widely, -omics approaches is particularly attractive in addressing the threats posed by and illuminating the biology of these pathogens. Typically, research into plant, human and wildlife mycoses have been largely separated, with limited dialogue between disciplines. However, many serious mycoses facing the world today have common traits irrespective of host species, such as plastic genomes; wide host ranges; large population sizes and an ability to persist outside the host. These commonalities mean that -omics approaches that have been productively applied in one sphere and may also provide important insights in others, where these approaches may have historically been underutilised. In this review, we consider the advances made with genomics approaches in the fields of plant pathology, human medicine and wildlife health and the progress made in linking genomes to other -omics datatypes and sets; we identify the current barriers to linking -omics approaches and how these are being underutilised in each field; and we consider how and which -omics methodologies it is most crucial to build capacity for in the near future.

## 1. Introduction

Emerging fungal pathogens and outbreaks of fungal disease pose serious yet often underappreciated threats to food security, wildlife populations, and human health. Globally, mycoses are known to directly cause serious or life threatening conditions in 150 million people, and infect over a billion [1,2,3,4]. Furthermore, the incidence of fungal pathogen outbreaks appears to be increasing [5,6,7,8], with mycoses more likely than any other class of pathogens to cause population declines, extirpations, or extinctions in their hosts when they occur [5]. However, assessing the true impact of fungal pathogens is challenging due to a lack of research investment and prioritisation [1,4]. Commanding less than 2% of the financial investment dedicated to infectious disease research in the UK, and less than 3% of funded studies [1,2,5], the morbidity and mortality which the fungal pathogens inflict upon host populations far outweighs the current research investment in their evolution, epidemiology, and mitigation.

The complexity of fungal host-pathogen interactions means that an -omics’ approach is suited to this research. In comparison to viral and bacterial pathogens, our understanding of fungal pathogen biology is poor (approximately 98% of fungi are not even described [9]), and the untargeted nature of -omics approaches makes them particularly valuable in expanding on current knowledge. Here, we use the term ‘-omics’ to mean high-throughput assays and analyses that comprehensively and simultaneously target molecules of a single type from a sample [10]. The scope of this expansive field cannot be covered in a single review, so here we largely focus on genomics, with some consideration of proteomics, metabolomics and transcriptomics. As -omics and high throughput investigative methods become more affordable, and the skills they require become more widespread in the research community, the potential for leaps in advances in understanding fungal pathogen biology has never been greater.

Fungal pathogens of any host often have common traits that may make them particularly able to capitalise on a globalised world, which makes them crucial candidates for increased research focus. Firstly, fungal pathogens tend to have wide host ranges. *Aspergillus* spp., are a ubiquitous group of environmental fungi, capable of causing a potentially fatal lung infection, invasive aspergillosis (IA), in susceptible human hosts. Beyond the human host, a small group of *Aspergillus* species are able to infect a staggeringly vast host range, including plants, insects, birds and mammals, most of which are common domesticated animals. The most common *Aspergillus* species to cause IA, *Aspergillus fumigatus*, is known to have a host range spanning honey bees, multiple bird species, dogs, cats, ruminants, horses, marine mammals, and monkeys, in addition to humans [11]. Secondly, fungi often have large, plastic genomes, capable of interspecific recombination and hybridisation; triticale, for example, is a wheat/rye hybrid crop that until recently was widely grown in large part due to its resistance to *Blumeria graminis*, the powdery mildew pathogen. Recently, however, a triticale-specialising powdery mildew hybrid pathogen, *B. graminis* f. sp. *triticale* has emerged, causing yield reductions of up to 60% in affected regions. The hybrid pathogen is the result of a cross between a wheat-specialising powdery mildew and a rye-specialising powdery mildew [12,13,14]. Finally, fungi are often able to persist in the environment without a host for considerable lengths of time. *Pseudogymnoascus destructans* (*Pd*) and *Batrachochytrium dendrobatidis* (*Bd*)*,* the causative agents of White Nose Syndrome (WNS) and chytridiomycosis, are both able to exist outside their hosts for extended periods of time and in the case of *Bd*, this ability has been linked to the level of extinction risk for the host [15,16]. This huge host range combined with common pathogenic traits make fungal pathogens ideal systems to study within the “One Health” framework and with an -omics approach. In this article, we compare how -omics approaches have been used to date to study outbreaks of fungal pathogens in the often-separated sectors of humans, plants and wildlife, and to highlight methods and approaches pioneered in some areas that may be underutilised in others (Figure 1). Infectious disease research would greatly benefit from an increase in cross-disciplinary approaches; where this has occurred already, translational benefits have been substantial. Increasing engagement and dialogue between researchers separated by their subject’s host taxonomy should be an urgent priority going forward [17].

## 2. -Omics and Fungal Pathogen Outbreaks in Crops

The lack of genetic diversity within crops of many agro-ecosystems, combined with high density agriculture over large spatial scales and the extensive regional specialisation of crop species creates conditions conducive to the rapid emergence and spread of fungal pathogens and the mycotoxins that they produce, threatening food security and causing substantial economic losses on a global scale [18,19]. Fungal pathogens in particular represent a major threat to the world’s crop supplies [20], and have precipitated famines in regions and crops as diverse as *Helminthosporium oryzae* destroying rice crops in Bengal in 1943 [21] to *Puccina* species which have caused devastating losses of many crops from the Roman ages through to the present day. In particularly severe cases, an entire harvest may be lost—an outbreak of rust in Ethiopia from 1993 to 1994 reduced yields by between 65% and 100%, causing severe food shortages for up to 300,000 people [22,23,24,25]. Fungal pathogens of crops are estimated to account for the loss of approximately 30% of global yields [26] and thus are a major contributor to the >USD 220 billion lost in crop yields to disease annually [27]. The severe consequences of fungal pathogen outbreaks in economic terms as well as in causing human morbidity and mortality mean that this is the area of research on mycoses where -omics has been applied most broadly to date.

Comparative population genomics is now commonly used to better understand the population structure of plant pathogenic fungi. As with other fields, the high-resolution insight provided by whole genome variant detection has afforded researchers with an unprecedented level of genetic markers to unravel the population dynamics of destructive crop pathogens [28]. Pathogens such as *Zymoseptoria tritici, Magnaporthe oryzae* and *Leptosphaeria maculans* have all had their genomes interrogated for clues into their recent and rapid adaptation to agricultural cops [29,30,31]. Comprehensive genome assembly algorithms have paved the way for large-scale predictive gene calling to further explore host-pathogen interactions; identifying gene candidates involved in novel virulent phenotypes that can be related back to disease events in the field [32].

A study investigating the genomics of wheat pathogen *Zymoseptoria tritici*, that emerged alongside the domestication of wheat sometime between 10,000 and 12,000 years ago, is an example of using comparative population genomics to historically explore the speciation of a novel plant pathogen [29,33,34]. Ultimately, it was shown that *Z. tritici* displayed signatures of increased adaptive evolution when compared to the ancestral species, and that there was evidence of positive selection, particularly on putative pathogen effectors, which can be considered a sign of the coevolutionary arms race between the pathogen and the host.

The physical structure of fungal genomes, together with the activity of transposable elements (TE), has been shown to play a role in the diversification of pathogen effectors, enabling species to avoid host-mediated defence strategies. Generating a high-quality genome assembly of oilseed pathogen *Leptosphaeria maculans*, helped elucidate the repertoire of effectors compartmentalised within adenine and thymine (AT)-rich repeat blocks [31]. These blocks, which have hallmark signatures of TE activity, also show evidence of repeat-induced point (RIP) mutations, a fungal-specific genome defence mechanism that halts the continuous activity of TEs. Small-secreted protein encoding genes (candidate effectors) located within close proximity to RIP activity may have been affected by a phenomenon known as RIP leakage [35], resulting in a high mutation rate that allows rapid diversifying selection to occur. These mutations, consequently, have the potential to rapidly create genetic variation within virulence factors which accelerate pathogen evolution.

One of the most recognisable applications of -omics in fungal plant pathology is through the improved understanding of disease dynamics and pathogen biology. Genome-wide methods, similar in their purpose to quantitative trait locus (QTL) analysis, are now being utilised to directly pinpoint variants associated with phenotypic traits, such as host defence mechanisms [36]. These methods have the latent potential to be used as target breeding strategies, exposing the molecular mechanisms driving host-pathogen interactions and exploiting them to counter the impact that rapidly emerging pathogens have on agricultural cropping yield. Examples include the detection of a previously unannotated avirulence effector (AvrStb6) in *Z. tritici*, which interacts with resistant wheat varieties harbouring the corresponding resistance gene (Stb6) [37]; and the identification of genomic regions with the potential discovery of novel blackleg resistance genes in oilseed rape varieties [38].

## 3. -Omics and Fungal Pathogen Outbreaks in Humans

Unlike crop mycoses, the impact of fungal infections on humans has only been widely acknowledged as important since the 1980s [39,40]. Out of 600 fungal species that are known to have caused infection in humans, only 30 regularly cause disease. However, these 30 species are responsible for over 300 million serious fungal infections each year across the globe [41], with 0.5% individuals dying from their fungal infection [4]. Whilst the increase in immunocompromised patients has prompted a rise in fungal infections, mycoses are also a public health concern for immunocompetent individuals [42,43]. Outbreaks of fungal infections can occur in healthcare settings, or during hospital construction and renovation [44], posing a serious threat to immunocompromised and hospitalised patients. The most common nosociomal fungal infection is *Aspergillus fumigatus*, causing a pulmonary infection. Recently, nosocomial outbreaks of *Candida auris*, a multi-drug resistant yeast, have occurred in many countries across the globe. In tropical and subtropical climates, fungi (such as *Cryptococcus* species) exist in ecological niches and are capable of creating outbreaks in the community amongst people who may or may not have predisposing medical conditions [45,46].

Genomics have already proved successful in the tracking of infection in bacterial and viral outbreaks [47,48,49]. The development of the Oxford Nanopore sequencing device, the MinION, has enabled the rapid ‘real-time’ sequencing of outbreaks. Given the successful use of the MinION on fungal pathogens [50,51] it is feasible that this cutting-edge technology could be applied to outbreaks of fungal pathogens to track the spread of infection and to aid in traditional epidemiological analyses. A logical extension of this is to use genomics to track and address existing and emerging drug resistance in pathogenic fungi; most genomics studies currently investigate known single nucleotide polymorphism (SNP)-causing substitution events, which confer drug resistance. However, many yeasts are known to undergo ploidy changes (e.g., *Candida* and *Cryptococcus* species [52,53,54]), and the application of transcriptomics can unlock our understanding of how changes in gene expression confer drug resistance.

Finally, future research should explore gene changes due to epigenetic changes, and how these may contribute to virulence and/or drug resistance; few studies to date have ventured into the field of fungal epigenomics, yet recent technological advances, such as Methyl C-seq on the Illumina HiSeq platform, should enable the exploration of epigenetic modifications more easily [55]. The most recent innovation in use the of -omics for managing mycosis outbreaks in humans has been the utilisation of rapid real-time sequencing technology such as MinION to track infection outbreaks in real time, quickly identify the population structure and possible routes of introduction of pathogens, and inform drug regimen and management decisions. This has been demonstrated in practice in two recent outbreaks of *Candida auris* in the UK.

## 4. -Omics and Fungal Pathogen Outbreaks in Wildlife

Somewhat surprisingly, some of the most high-profile fungal pathogen outbreaks in recent times have occurred in wildlife populations, whereas the devastating impacts mycoses are capable of inflicting on host populations has been brutally demonstrated despite probably the lowest level of research and surveillance investment of all three sectors discussed here. *Bd* and *Batrachochytrium salamandrivorans* (*Bsal*), which cause the skin disease chytridiomycosis in amphibians, have caused extinctions of up to 500 species and *Pd*, the causative agent of WNS in bats, continues to cause die-offs and population extirpations of previously common bat species across North America [56,57]. The vast majority of -omics based research in this field to date has been applied to these three pathogens, but the recent emergence of a novel mycosis in snakes, caused by *Ophidiomyces ophiodiicola*, and *Aspergillus* infections in endangered bird species in New Zealand [58,59], present opportunities to apply the technological progress made in research chytridiomycosis and WNS to novel systems, with the aim of improved wildlife conservation outcomes, faster.

Comparative genomic and transcriptomic studies of *Bd* and *Bsal* represented a huge leap forward in how fungal pathogens of wildlife are studied. The substantial levels of funding required for these research projects was made possible by the uniquely destructive nature of these diseases at a time when pathogens of wildlife, and fungi in particular, were not generally considered of general concern. Comparative genomics of *Bd* has revealed the existence of multiple lineages within the species, which were subsequently found to have varying levels of virulence and distinct distributions around the world [60,61,62]. Comparative genomics of chytrids has enabled the identification of gene families associated with pathogenicity and evidence of divergent infection strategies of *Bd* and *Bsal* [63]. Meanwhile, transcriptomics has revealed how host responses to infection vary depending on the host species but also the lineage of fungus infecting [64,65], and that the fungus itself exhibits host-specific gene expression [66]. Further emphasising how -omics reveals the extreme complexity of fungal host-pathogen systems, recent research has shown that environmental conditions such as temperature also mediate the transcriptional host response to infection [67]. The most recent advances have also addressed the microbiome of the host, drawing correlations between the microbiome, the metabolome and the transcriptome of the microbiome with disease status [68,69,70,71], with the hope of using this information to target mitigation strategies [72].

The huge research interest in *Bd* emergence meant that when *Pd* emerged in North American bats, the integration of -omics into research focusing on the pathogen occurred at a much faster pace. Following its discovery in 2006 [56,73] and description in 2009 [73], multiple studies have already reported analyses of the host microbiome, comparative genomic analyses and host-specific transcriptomic responses to infection. Whether this rapidity can be increased again in response to other more recently emerging fungal pathogens remains to be seen; Snake Fungal Disease (SFD), caused by *Ophidiomyces ophiodiicola* (*Oo*), was first recorded in 2006 as a syndrome of unknown aetiology causing anorexia, lesions and poor body conditions in threatened wild snakes primarily in North America [59,74]. Despite the lack of any known treatment, genomic approaches to this emerging pathogen have been underexplored to date. However, the first *Oo* genome was published in 2017, and -omics approaches hold the potential to rapidly increase our understanding of this emerging pathogen. In the case of SFD, the power and utility of genomics must be considered alongside the limited funding available, with the investment heavy technology being deployed where it can be of maximum utility for the smallest cost.

Despite impressive advances in the field of emerging fungal pathogens of wildlife since the emergence of *Bd* in 1999, progress is hampered by a lack of financial investment. In turn, this impacts the quality and breadth of online resources available to support -omics work, such as genome databases and the availability of published annotated genomes. As a result, despite wildlife pathogen systems being far more amenable to an experimental approach informed by -omics than human fungal pathosystems, most studies remain correlative rather than investigating causation, unlike in plants where experimental work is pushing frontiers. This lack of research investment is short-sighted, as evidenced by the catastrophic impacts inflicted on wildlife populations, in a very short space of time, by WNS and chytridiomycosis. For progress across all fields, in crop health and human health as well, a focused effort to apply a sophisticated -omics approach to researching wildlife pathogens could lead to great cross-disciplinary benefits and crucial insights into how the most dangerous mycoses known have evolved and interact with their hosts and environment.

## 5. Discussion

The plant pathology sector continues to be at the forefront of the field in terms of applying -omics approaches to understand and tackle emerging mycoses. In some cases, such as exciting avenues like engineering plant resistance to fungal pathogens, this is due to not only to a higher commercial interest, but also reduced ethical barriers for experimentation with crop systems. Advances of this nature would be highly challenging and likely unethical for translation into wildlife or human research. However, all three sectors stand to benefit from exchanging translatable approaches and methods. For example, significant progress in recent years in plant pathology has been made using knock-out experiments, which have been applied only in a limited manner in human–fungal and wildlife-fungal pathogen systems. Conducting gene-knockout experiments on the pathogens themselves may go a long way to illuminating mechanisms of infection and key virulence factors in these organisms. Similarly, all three areas are on the brink of a more thorough exploration of metagenomics to approach a more holistic understanding of pathogen and microbial communities, which may shed light on disease ecology and thus the conditions under which disease is most likely to emerge.

Several high-profile drug-resistant mycoses have emerged in recent years, triggering an urgency that has resulted in several rapid advances in the use of -omics approaches in human-fungal pathogen systems. A key priority in the near future is to utilise genomics to identify genome regions conferring drug resistance, and to develop methods that would allow the surveillance of hospital patients for the emergence of not only fungal pathogens, but drug resistance within those fungal pathogens. It goes without saying that -omics technologies are proving key to developing the next generations of antifungal drugs that are needed now and in the future [26].

The application of -omics techniques developed for analysing ancient DNA (aDNA) is to date underutilised across the majority of research into contemporary pathogen outbreaks. The highly sensitive nature of methods developed for use with aDNA would be invaluable in clinical settings where sampling may not be optimal, or where the target pathogen may be at a low concentration within a large and complex microbial community, and thus hard to detect soon after infection [75].

Additionally, the incorporation of information on historical pathogen outbreaks may provide new insights into contemporary disease ecology. Paleoepidemiological investigations of historical outbreaks using aDNA have benefited from the additional genomic data, significantly enhancing the understanding of the origin of pathogenicity and evolutionary history [76]. For example, the reconstruction of ancient genomes of plague bacterium, *Yersinia pestis,* clarified the phylogenetic relationships between the widely known Black Death outbreak in Europe (14–17th centuries) to the descendant strain responsible for the ‘third plague pandemic’ confined to Asia (19–20th centuries) [77], and furthermore, to the ancestral strain which caused Plague of Justinian (6–8th centuries) [78].

While dating phylogenetic trees can be described with modern genomes through estimations of both the timing of evolutionary events, and molecular evolution, aDNA represents a unique source of information that modern estimates cannot contribute [79]. Foremost, taxonomy is highly dynamic through space and time, and the complexities of such histories can only be reconstructed from extant modern descendants; aDNA genomic data can provide information of local extinctions, migrations, and admixture events that are not apparent in modern DNA [80]. Furthermore, fungal communities are interactive with the ambient environment, and anthropogenic processes of the past century may induce selective pressures upon extant communities [81], confounding phylogenetic inferences.

We believe that the key barriers to fully exploiting -omics approaches in the study of fungal pathogens are a lack of investment in surveillance and funding available for -omics approaches, and the practical challenges associated with integrating multiple -omics data forms. Mycosal epidemiology is vastly underfunded in comparison to viral and bacterial diseases, and the expensive nature of many -omics approaches means that these methods are simply out of reach for many researchers. However, the prices associated with this research continue to fall rapidly, and in concert the breadth and depth of analysis expertise among mycotic researchers increases. As such, it seems hopeful that these barriers will soon be overcome. Similarly, the issue of multi-omics data integration and analysis is increasingly recognised as a barrier to progress, and promising tools are being developed to address this issue [82,83] More pressingly, much -omics work relies on having access to comprehensive databases, and fungal genomes are severely underrepresented. Meeting this need will require concerted effort and collaboration among researchers in all three sectors outlined here. Building these databases will benefit the investigation of mycoses in all hosts, and it is crucial that the data obtained through -omics studies continues the trend of being provided in an easily accessible manner, and free for other researchers. Doing so will enable the full exploitation of powerful -omics approaches in tackling the urgent issue of emerging fungal pathogens, pushing wildlife, crop and human health research forward in concert.

## 6. Case Studies

### 6.1. Case Study 1: Magnaporthe Oryzae Genome Reconstruction and Proteomics Yield Insights into Pathogen Biology and Infection Process

*M. oryzae* is a destructive seed-borne pathogen of rice (*Oryzae sativa*) and one of the most economically devastating crop diseases globally [84]. Typical crop losses range between 10% and 30%, and localised outbreaks can have a major influence on crop quality and yield [85,86]. *M. oryzae*, and its coevolution with rice, is now a well established model system for conventional host-pathogen interactions [87], where traditional molecular methods are increasingly supported by -omics-based approaches, including comparative and functional genomics, transcriptomics and proteomics. The *M. oryzae* genome was first sequenced in 2005 (strain 7–15), notable as the first plant pathogen to have its genetic sequence resolved [88]. The reconstruction of the genome, coupled with the pathogen’s experimental plasticity, has helped clarify key components of the *M. oryzae* disease cycle and drivers behind its pathogenicity. Insights, such as an expanded family of G-protein signalling receptors, which play a critical role in appressoria formation, and the evidence of transposable element activity affecting genome structure, have guided further exploratory -omics. Differential transcriptomics, comparing wild-type and knock-out strains (Guy11 and Δ*pmk*1) during an artificial infection, revealed transcripts necessary for invasion, particularly Pnk1 MAP kinase as a global regulator of appressorium-associated gene expression [89,90]. Proteomics studies have helped elucidate the function of *M. oryzae*’s repertoire of secreted peptides roles such as cell wall modification, reactive oxygen species detoxification; lipid modification, metabolism, and protein modification, all have prospective roles in leaf invasion and infection [91,92,93]. Foundational methodologies such as these have led to more novel techniques, utilising the upwards trend of DNA sequencing capabilities. Techniques such as genome-wide association studies, large-scale population genomics and high-quality genome assemblies and annotations will pave the way for breeding strategies that will assist in the deployment of resistance-associated genes across a huge range of cultivated plants, reducing the overall impact fungi have an agro-systems worldwide, strengthening food security for vast swathes of the human population. 

### 6.2. Case Study 2: Genomics to Inform Management Candida Auris Nosociomal Infection Outbreaks in the UK

*C. auris,* first isolated in 2009 from the ear canal of a hospitalised patient in Japan [94], is now a globally emerging multi-drug resistant nosociomal pathogen [50,95]. Genomic analyses have been used in several Candidiasis outbreaks to identify sources of infection and map the spatial and temporal spread of the pathogen. In 2018, rapid whole genome sequencing technology was utilised to retrospectively analyse the largest outbreak of *Candida auris* in the UK to date, at an inner London hospital. Using MinION technology, researchers were able to determine that multiple sources of infection were unlikely; that the most recent common ancestor of the three outbreak clades recovered from patients dated to March 2015; and crucially were able to identify which isolates had reduced susceptibility to specific fungicides. This study represented the first use of a MinION sequences on a human fungal pathogen, and demonstrated how the ability to generate long reads rapidly made this technology of particular utility to fungal pathogen outbreak scenarios [50]. The ability of genomic analyses to inform outbreak management was further demonstrated in another UK *C. auris* outbreak in an intensive care unit outbreak in Oxford, UK. Here, the detection of *C. auris* isolates on reusable equipment, but rarely in the general environment, that were closely related to those recovered from patients, resulted in a management decision to remove reusable temperature probes which was followed by a reduction in patient infections [96].

### 6.3. Case Study 3: Population Genomics and Phylogenomics to Identify the Causal Agent, Emergence and Dispersal of a Cryptococcosis Outbreak in the Pacific North West

The species complexes *Cryptococcus gattii* and *Cryptococcus neoformans* occur ubiquitously in the environment and can cause cryptococcosis when infectious airborne propagules are inhaled by a susceptible individual. Whereas *C. neoformans* infection is more commonly associated with a highly fatal acute meningitis, particularly among those who are immunocompromised, *C. gattii* infection more commonly manifests as a pneumonia and affects both immunocompromised and immunocompetent individuals [97]. Historically, *C. gatti* was thought to be restricted to tropical and subtropical regions [98], however, in 1999 *C. gattii* was identified as the causal agent in an outbreak of cryptococcosis in British Colombia, Canada, and the Pacific Northwest (PNW), representing the first outbreak of the pathogen in a temperate zone [99]. Genomics approaches were subsequently used to understand the outbreak in three key ways. Firstly, population genomics was used to identify the causal outbreak lineages and identified two discrete genotypes of VGII from both clinical and environmental sources—VGIIa (major) and VGIIb (minor)—representing two independent clonal populations, as well as a third novel genotype (VGIIc) [98,100]. Subsequent whole-genome sequence typing (WGST) comparing the major and minor genotypes to one another and to global VGI identified differences in chromosome copy numbers, genomic arrangements and a large number of sequence polymorphisms [101,102].

Secondly, phylogenomics identified the origins of the outbreak, including from where and when lineages were introduced, as well as how the lineages subsequently spread from Vancouver across British Colombia and PNW. Analysis suggested the major outbreak lineage, VGIIa, had originated in South America and spread via multiple dispersal events to North America, while VGIIb originated in Australia and was likely independently introduced. Both lineages showed evidence of subsequent clonal expansion in PNW [103,104] as well as use of same-sex mating to expand its geographical range [100].

Finally, genomics analyses were used to understand the emergence and biology of virulence in these lineages. The outbreak strains demonstrated hypervirulence in comparison to global isolates [105,106]. Through phylogenomics, the hypervirulent phenotype seen in VGIIa was found to emerge from a less virulent ancestor through a transitory mutator phenotype [104]. Through comparative whole genome analysis, the hypervirulent VGIIa was found to have numerus differences in chromosome copy number, genomic rearrangements and sequence polymorphisms compared to global VGI, and gene content comparisons found genes potentially related to habitat adaptation, virulence and clinical outcomes [103]. Virulence differences even between highly related strains of VGIIc were also explored and identified a small number of single site variants within coding regions thought to contribute to more virulent phenotypes [107].

### 6.4. Case Study 4: Novel -Omics-Based Diagnostics and Point of Origin Investigations for Aspergillus Fumigatus Infection in Kākāpō

The Kākāpō are a critically endangered bird species endemic to New Zealand [108]. Their charismatic nature has driven widespread interest and popular support for the conservation of the species, and in 2010, a Kākāpō named Sirocco was even named New Zealand’s “Official Spokesbird for Conservation” by the then Prime Minister, John Key. In 2019, an outbreak of aspergillosis affected 21 birds (over 10% of the global population) and resulted in the deaths of nine individuals [109]. The management of this outbreak, while ongoing, has relied heavily on -omics approaches. Genomic analyses of *A. fumigatus* isolates recovered from infected birds, their nests, and food sources are being used to indicate how closely related the infecting isolates are, whether they come from the same source, and what that source is likely to be; all crucial information for developing conservation action plans. The rapid diagnosis of aspergillosis is critical for prompt treatment and to improve infection outcomes, but many diagnostics that are routine in humans are not applicable to wildlife as they would require sedation. A non-invasive novel diagnostic based on metabolomics, the eNose, has recently been developed to detect the volatile organic compounds (VOCs) produced by *A. fumigatus* in the breath of infected individuals with 89% accuracy [110]. Similarly, the detection of exhaled VOCs by gas chromatography–mass spectrometry has reported 94% sensitivity and 93% specificity in detecting aspergillosis [111]. In falcons, nuclear magnetic resonance performed on biological samples such as blood and serum is capable of detecting changes in the metabolome between healthy and infected individuals and has been used to detect aspergillosis [112]. Novel tools such as this, grounded in -omics, present exciting opportunities to approach wildlife pathogen diagnostics in new, specific and non-invasive ways.

### 6.5. Case Study 5: Investigation into Host Defences against Infection by Batrachochytrium Dendrobatidis (Bd) via a Metabolomics Approach

*Bd*, which causes the potentially lethal disease chytridiomycosis, has precipitated unprecedented declines in hundreds of amphibian populations globally [57]. Comparative genomic analyses have revealed the existence of multiple epidemiologically relevant lineages within the species [61,62,113,114]; enabled the identification of virulence factors associated with pathogenicity among the chytrid fungi [62,115]; and enabled the point of origin and global dispersal patterns of *Bd* lineages to be identified [60]. Transcriptomics have revealed the gene expression profiles associated with different life stages of the pathogen [116] and, more recently, metabolomics is illuminating host defence mechanisms. Amphibians and their skin microbiota produce a wide range of secondary metabolites with antimicrobial activity, presenting a complex physical barrier to pathogens; differences in host metabolome correlate with *Bd* infection status [117]. Recent advances in gas chromatography, liquid chromatography and mass spectrometry have allowed the characterisation and quantifications of the complex chemical mixtures in the amphibian skin metabolome and thus allowed the relationship between metabolome composition and anti-microbial activity to be established via in vitro assays [118]. However, the processes which shape the amphibian metabolome, and therefore disease resistance, while known to be modulated by environmental stressors, remain poorly understood.

Applying techniques developed for studying phenotypic plasticity in plants could greatly increase our understanding of the heterogeneity observed in natural amphibian–*Bd* systems [119] and interaction with anthropogenic influences [120]. Further exploration of the amphibian metabolome may also allow the exploration of the role of the metabolome in host adaptation and the implications of environmental change on the epizootic [121].

## Figures and Tables

**Figure 1 life-10-00315-f001:**
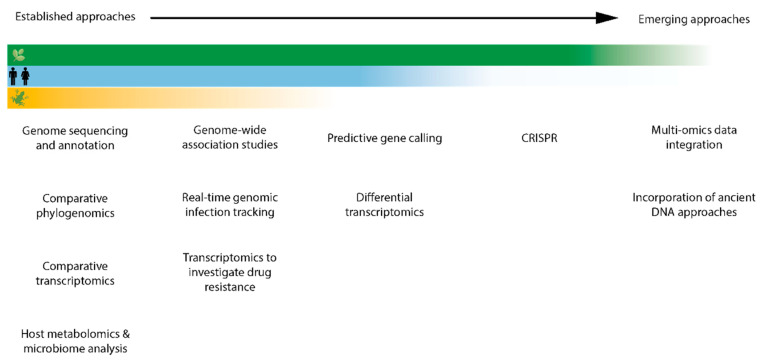
Graphical representation of progress across crop (**green**), human (**blue**), and wildlife (**yellow**) fungal pathogen research in applying established and emerging -omics approaches. The extent to which each sector makes use of each approach is indicated by the intensity of colour.

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
