# Peer review of "Cross-Disciplinary Genomics Approaches to Studying Emerging Fungal Infections"

_life, 2020, doi:10.3390/life10120315_

Round 1
Reviewer 1 Report
This review article describes the use of omics approaches to study fungal pathogens relevant to plant, human and wildlife health.
The overall paper is well written. The approach of reviewing the fungal pathogen research across field- plant, human and wildlife- is interesting and useful.
Suggestions:
1) The paper largely focusses on genomics approaches and studies with brief mentions of other omics such as transcriptomics, proteomics and metabolomics in the case studies.
I recommend that the authors either change the title and wording of the introduction to reflect that this is a review primarily on genomics approaches- which would require few edits to the current paper- or alternatively include more studies and references of other omics approaches, especially mass-spectrometry based omics such as metabolomics, lipidomics and proteomics.
A lot of recent work has used lipidomics to identify antifungal drug targets and therapeutic interventions.
2) When studies use multiple omics approaches the huge challenge lies in integration of the various omics data types. This is something that should be mentioned as a barrier in interpreting omics data.
3) In line 116 on page 3, what does the "physical nature" of fungal genomes mean? Please check if this is a typo.
Author Response
We would like to thank the reviewer for the positive response and for the useful suggestions. We have addressed the suggestions as below:
1) The paper largely focusses on genomics approaches and studies with brief mentions of other omics such as transcriptomics, proteomics and metabolomics in the case studies.
I recommend that the authors either change the title and wording of the introduction to reflect that this is a review primarily on genomics approaches- which would require few edits to the current paper- or alternatively include more studies and references of other omics approaches, especially mass-spectrometry based omics such as metabolomics, lipidomics and proteomics.
A lot of recent work has used lipidomics to identify antifungal drug targets and therapeutic interventions.
Response: We agreed with the reviewer that our review was heavily focussed on genomics. We felt that as the majority of our expertise lay in this area, it would be better to change the title and wording of the introduction, as suggested. The title is now “Cross-disciplinary genomics approaches to studying fungal infections”, and we have updated both the abstract and introduction to reflect this change.
2) When studies use multiple omics approaches the huge challenge lies in integration of the various omics data types. This is something that should be mentioned as a barrier in interpreting omics data.
Response: We have updated our discussion to include the integration of various omics data types as a barrier, and included this aspect in the new figure (Figure 1), as an approach that needs to be fully addressed by all three sectors discussed in the review. The relevant section of the discussion now reads as follows:
“We believe that the key barriers to fully exploiting -omics approaches in the study of fungal pathogens are a lack of investment in surveillance and funding available for -omics approaches, and the practical challenges associated with integrating multiple -omics data forms. Mycosal epidemiology is vastly underfunded in comparison to viral and bacterial diseases, and the expensive nature of many -omics approaches means that these methods are simply out of reach for many researchers. However, prices associated with this research continue to fall rapidly, and in concert the breadth and depth of analysis expertise among mycotic researchers increases. As such, it seems hopeful that these barriers will soon be overcome. Similarly, the issue of multi-omics data integration and analysis is increasingly recognised as a barrier to progress, and promising tools are being developed to address this issue [82], [83] More pressingly, much -omics work relies on having access to comprehensive databases, and fungal genomes are severely under-represented. Meeting this need will require concerted effort and collaboration among researchers in all three sectors outlined here. Building these databases will benefit the investigation of mycoses in all hosts, and it is crucial that data obtained through -omics studies continues the trend of being provided in an easily accessible manner, and free for other researchers. Doing so will enable the full exploitation of powerful -omics approaches in tackling the urgent issue of emerging fungal pathogens, pushing wildlife, crop and human health research forward in concert.”
3) In line 116 on page 3, what does the "physical nature" of fungal genomes mean? Please check if this is a typo.
Response: Thank you for spotting this, we have updated “physical nature” to “physical structure”.
Reviewer 2 Report
The manuscript entitled "Cross-disciplinary -omics approaches to studying emerging fungal infections" reviews the fungal infections and the latest -omic thecnologies that are being used in the field.
In my opinion, the manuscript is suitable to be published in Life after minor revisions.
The introduction is well written, the case studies presented are important issues.
Comments:
- Abbreviations should be described when used for the first time
- The review should have a figure focusing the importance of each topic and the information retrieved from each subject
- The case studies should be presented diferently, it is confusing in the same section
Author Response
We would like to thank the reviewer for the positive response and for the useful suggestions. We have addressed the suggestions as below:
- Abbreviations should be described when used for the first time
Response: Thank you for spotting these errors – we have double checked the manuscript and ensured that all abbreviations have been described at the first use.
- The review should have a figure focussing the importance of each topic and the information retrieved from each subject
Response: We have now included a figure (Figure 1) summarising the review. The figure depicts how many established and emerging approaches each of the three sectors of crop research, human research, and wildlife research are using.
- The case studies should be presented differently. It is confusing in the same section
Response: We have now moved the case studies to their own section at the end of the discussion.